# Performance Characteristics of a Novel 3D-Printed Bubble Intermittent Mandatory Ventilator (B-IMV) for Adult Pulmonary Support

**DOI:** 10.3390/bioengineering9040151

**Published:** 2022-04-02

**Authors:** Jonathan A. Poli, Christopher Howard, Alfredo J. Garcia, Don Remboski, Peter B. Littlewood, John P. Kress, Narayanan Kasthuri, Alia Comai, Kiran Soni, Philip Kennedy, John Ogger, Robert M. DiBlasi

**Affiliations:** 1Center for Integrative Brain Research, Seattle Children’s Research Institute, Seattle, WA 98101, USA; robert.diblasi@seattlechildrens.org; 2Medical Sensor Systems, Seattle, WA 98126, USA; 3Division of the Biological Sciences, University of Chicago, Chicago, IL 60637, USA; ajgarcia3@uchicago.edu (A.J.G.III); littlewood@uchicago.edu (P.B.L.); jkress@medicine.bsd.uchicago.edu (J.P.K.); bobby.kasthuri@gmail.com (N.K.); 4Neapco Holdings, LLC, Farmington Hills, MI 48336, USA; dremboski@neapco.com (D.R.); acomai@neapco.com (A.C.); ksoni@neapco.com (K.S.); pkennedy@neapco.com (P.K.); jogger@neapco.com (J.O.); 5Department of Respiratory Care, Seattle Children’s Hospital, Seattle, WA 98105, USA

**Keywords:** bubble continuous airway pressure, ventilator, additive manufacturing

## Abstract

The COVID-19 pandemic has brought attention to the need for developing effective respiratory support that can be rapidly implemented during critical surge capacity scenarios in healthcare settings. Lung support with bubble continuous positive airway pressure (B-CPAP) is a well-established therapeutic approach for supporting neonatal patients. However, the effectiveness of B-CPAP in larger pediatric and adult patients has not been addressed. Using similar principles of B-CPAP pressure generation, application of intermittent positive pressure inflations above CPAP could support gas exchange and high work of breathing levels in larger patients experiencing more severe forms of respiratory failure. This report describes the design and performance characteristics of the BubbleVent, a novel 3D-printed valve system that combined with commonly found tubes, hoses, and connectors can provide intermittent mandatory ventilation (IMV) suitable for adult mechanical ventilation without direct electrification. Testing of the BubbleVent was performed on a passive adult test lung model and compared with a critical care ventilator commonly used in tertiary care centers. The BubbleVent was shown to deliver stable PIP and PEEP levels, as well as timing control of breath delivery that was comparable with a critical care ventilator.

## 1. Introduction

The COVID-19 pandemic caused by the novel corona virus (SARS-CoV-2) has led to over five million deaths worldwide at the time of writing, with the global weekly death toll remaining in the thousands [1]. In the worst cases, COVID-19 patients exhibit a form of severe acute respiratory distress syndrome (ARDS) that requires mechanical ventilation to stabilize gas exchange while minimizing respiratory effort [2,3].

Early during the COVID-19 pandemic, it was projected that the need for mechanical ventilation would vastly exceed the number of mechanical ventilator devices available during hospital surge capacity [4]. This anticipated need led to substantial and widespread efforts to develop and manufacture devices for emergency respiratory support. Fortunately, a global shortage in patient ventilators has not yet emerged; the combination of effective public health policies and the rapid evolution in standard of care management strategies for COVID-19 patients has prevented an overwhelming strain of healthcare ventilator resources in many countries. For example, respiratory support provided by continuous high-flow nasal cannula [5] and helmet continuous positive airway pressure (CPAP) [6] have emerged as treatments for hypoxemic COVID-19 patients, thus mitigating the need for elective mechanical ventilation.

Despite progress toward effective respiratory support for combating COVID-19, elective intubation and invasive ventilation continues to be a last resort option for COVID-19-induced ARDS. Thus, as the pandemic continues, the concern that demand for effective respiratory support devices will outstrip availability remains a very real possibility, particularly in low-to-middle-income countries (LMICs) where access to mechanical ventilation is limited [7]. An inexpensive and reliable technology is needed to address this concern.

A multitude of low-cost, scalable solutions have been published since the onset of the pandemic in response to the sharp increase in demand for respiratory support technology [8]. Many of these solutions leverage designs that integrate existing respiratory support devices to generate manual or non-synchronized, automated breaths. Many of these solutions also offer positive inspiratory pressure (PIP) and positive end-expiratory pressure (PEEP) with intermittent mandatory ventilation (IMV) modes. While there are many ways to create the pressure differential (∆P) needed to generate inspiratory tidal volume (V_T_) to support alveolar minute ventilation, Mora et al. summarized the designs into three categories: ambulatory bag-mask unit (AMBU) bag compressor, blower, and pressure regulator designs [9]. AMBU bag compressor designs leverage widely available inexpensive manual resuscitators and PEEP valves, commonly used during CPR and transport, as the source of pressure differential needed by mechanically automating the frequency and PIP to prevent prologued manual inflation by clinicians. Design proposals such as GlasVent [10], MADVent [11], AIR ventilator [12], and the HDvent [13] are examples of AMBU bag compressor technologies. Blower designs control the breaths delivered to a patient by controlling the rotational spin and duration of a blower fan to produce intermittent pressure fluctuations. Khan et al. developed a helmet-based noninvasive ventilator (NIV) leveraging blower technology [14]. Pressure regulator designs that use a high-pressure gas source and timed solenoid-reducing valves to deliver PPV breaths include the OneBreath ventilator [15], the Portsmouth ventilator [16], and the PVP1 ventilator [17]. The performance and cost of these published, open-source designs attests to the ingenuity of the makers. However, these designs need electronic controllers and electricity to drive the pressure generating mechanism, which is a potential setback in limited resource and surge capacity settings where supply chains are disrupted, and grid power and batteries may be unavailable.

A class of neonatal noninvasive respiratory support devices [18,19] employing bubble continuous positive airway pressure (B-CPAP) via nasal prongs or mask is used to support pre-term infants with mild–moderate respiratory distress in LMICs [20]. The effectiveness of B-CPAP in larger pediatric and adult patients is technically challenging using existing low-flow devices. While B-CPAP clinical principles initially described by Wung et al. [21] focused on generating constant pressure via underwater seal and applying it to nasal airway openings of spontaneously breathing pre-term infants. A large fraction (60–80%) of pre-term newborns are not able to be supported with B-CPAP due to apnea and poor gas exchange. The introduction of the Hansen ventilator (HansenVent) was the first attempt at utilizing a device capable of generating CPAP and bubble IMV (B-IMV) to provide either invasive or noninvasive PPV. The HansenVent showed greater ventilation efficiency than a critical care ventilator in a neonatal animal model of severe respiratory insufficiency [22]. Recently, the Neovent demonstrated the feasibility of B-IMV for infant airway support without the use of electricity [23,24,25]. Despite this advance and the recent trials using B-CPAP in adults [26], little progress has been made to develop technology that effectively employs bubble-based airway support for both B-CPAP and B-IMV to support varying disease severity in adults. The use of the hydro-pneumatic principles found in B-CPAP to generate intermittent positive pressure inflations (PPV) to support gas exchange and offset high intrinsic work of breathing (WOB) levels in adult patients experiencing moderate to severe forms of respiratory failure appears unstudied.

The BubbleVent is a novel respiratory support apparatus based on the hydro-pneumatic principles of B-CPAP. Hydro-pneumatics in this case can be summarized as the use of an underwater seal at a known, controllable depth to generate inspiratory and expiratory pressure with stochastic pressure oscillations. The current device configuration is designed for large pediatric or adult use, but neonate and small pediatric configurations are also possible. The apparatus is currently composed of 3D-printed expiratory valve components combined with commonly available tubes, hoses, and tubing connectors to create an air-powered, hydro-pneumatic PPV generator. The BubbleVent is also suitable for manufacture using an injection molding process. The expiratory valve includes a breathing circuit valve (BCV) that allows cycling between independent inspiratory and expiratory tubes submerged within water seals to provide adjustable PIP/PEEP suitable for fully supported mechanical breath types. Deactivation of the BCV also allows the BubbleVent to deliver CPAP in spontaneously breathing individuals. The ability of BubbleVent to control both PIP and PEEP, inspiratory time (T_I_), and breathing frequency makes this a complete critical care ventilator that operates without direct electricity.

The purpose of the current study is to assess the performance and capabilities of the BubbleVent across a range of PIP, PEEP, bias flows, T_I_, and respiratory rate (RR) settings commonly used in adults with acute respiratory ARDS in the critical care setting. We hypothesized that the T_I_, RR, PIP, PEEP, and V_T_ delivered to a mechanical model configured with normal lung mechanics and mild–severe ARDS disease are not different between the BubbleVent and a critical care ventilator.

## 2. Materials and Methods

### 2.1. Description and Operation of the BubbleVent

The BubbleVent is a hydropneumatic PPV-generating apparatus positioned downstream of both the breathing gas source and the simulated patient (Figure 1). While the BubbleVent employs principles such as those used with a B-CPAP system [27], the BubbleVent can achieve time-cycle, pressure-limited IMV breaths using a single, constant gas source; this single gas source provides inspiratory flow, controls the BCV, and generates PIP and PEEP. This gas source can be a compressor, air blower, or air pump. The BubbleVent is designed for use with a range of breathing gases, such as room air, oxygen, or nitric oxide. The BubbleVent is designed to be incorporated with existing clinical devices such as flow meters, breathing gas blenders, a gas heater and humidifier, and a range of patient interfaces such as endotracheal tubes and bi-nasal prongs. Inspiratory flow can be set between 10–80 L per minute (LPM). Unlike conventional pressure-limited breath types, the gas bubbling through water creates small-amplitude high-frequency oscillations that have previously been shown to be beneficial for enhancing lung volume and gas exchange in surfactant-deficient animal models [19,22].

#### 2.1.1. Airway Pressure Control

The breathing gas pathway moves gas through the inhalation limb of the breathing circuit and patient interface in a continuous unidirectional bias flow. Gas then enters the exhalation limb passing through and exiting the system either through the PIP or PEEP outlet tubes. The exit path is regulated by opening and closing the BCV. PEEP is provided when the BCV is open, or the BCV is deactivated. PIP is provided when the BCV is closed (Figure 1). The PIP and PEEP tubes are placed in separate water tanks. This separation allows the water levels of each compartment, and consequently the intended PIP and PEEP levels, to be independently controlled (Figure 2).

#### 2.1.2. Control of PIP and PEEP Delivery

The central component of the BubbleVent is a vertical, poppet-style valve head (Figure 2a). The state of the valve head (up/down) controls the directional characteristics of the gas flow by blocking the flow of gas through the BCV (Figure 1, Table 1). The state of the valve head is controlled by the float via a mechanical linkage. This linkage means a change in the float state (up or down) directly controls the valve head state (down or up). This coupling also means the timing of each valve state is determined by the rate at which the float rises (float up) and falls (float down) (Figure 1, Table 1).

PIP and PEEP pressures are controlled independently by varying the water level in the PIP and PEEP tanks, the higher the water level in the tanks, the greater the PIP and PEEP pressures.

#### 2.1.3. Control of PIP and PEEP Timing

*BCV Open to Closed*: To deliver PIP, the BCV must close. To close the BCV, the float must rise. To cause the float to rise, the float air chamber must become buoyant.

To fill the float air chamber, air is delivered directly to the submerged float air chamber via the float gas flow tube. Gas is provided by a low-flow air source with an adjustable flow rate (0–20 LPM); as the volume of gas required is small, air can be drawn directly from the inspiratory gas source. As the float air chamber fills with air, the float water in the float air chamber is displaced causing the float to become buoyant; a latch mechanism (PEEP timing control) ensures the float air chamber is full before the float can rise. Once the latch is triggered the float rises (Figure 1b—float up), causing the BCV to close quickly. Closure of the BCV directs the breathing gas to the PIP tube (Figure 1b).

*BCV Closed to Open*: To deliver PEEP, the BCV must open. To open the BCV the float must fall. To cause the float to fall the float air chamber must sink.

As the BCV closes, the float valve button is pressed simultaneously and opens the float valve (Figure 1b). Air exits the float air chamber via the float valve gas outlet, and the float chamber begins to fill with water. The rate at which gas passes through the float valve gas outlet is controlled using a resistor in the form of a ball valve, and this rate controls the rate at which buoyancy is lost; a latch mechanism (PIP timing control) ensures the float air chamber is empty before the float can sink. Once the latch is triggered the float sinks (Figure 1a—float down), causing the BCV to open quickly. As the PIP pressure is greater than the PEEP pressure, opening the BCV allows the breathing gas to find the path of least resistance and exit through the PEEP tube (Figure 1a).

*In summary*: The timing of the BCV open/close cycle is adjusted by independently varying the rates of the gas filling and exiting the float. Varying these rates changes the time it takes to fill and empty the float, which also determines the time the BCV is open or closed. In other words, varying the rates of air flowing into and out of the float varies the expiratory and inspiratory times (T_E_ and T_I_).

#### 2.1.4. Noninvasive Monitoring with Sequoia

A complete critical care ventilator requires an interface to monitoring its performance. Data about the performance of the BubbleVent were collected using Sequoia (Medical Sensor Systems, Seattle, WA, USA). Sequoia is a low-cost sensor array platform designed to monitor physical variables in the healthcare setting. Sequoia is capable of sampling, synchronizing, processing, and storing up to 6 different real-time signals sampled at 100 Hz. Sequoia was adapted to support airway pressure (P_AW_), bias flow, RR, and T_I_ monitoring of the BubbleVent. This adaptation of Sequoia is termed the BubbleVent Monitoring System (BVMS). The data was cleaned and restructured before being passed to a personal computer via USB. Sampling rate of the system was at 100 Hz. Details about the sensor array can be found in the Appendix A.

### 2.2. BubbleVent Manufacturing

The design process for the BubbleVent followed the prototype phase of quality management system (QMS) used by the medical sensor systems.

Functional requirements for the BubbleVent focused on accessibility, versatility, and capability. To support accessibility, the BubbleVent is designed to be inexpensive, simple to make and assemble, and easy to operate safely. The BubbleVent is also designed to be produced using both a range of 3D printing processes and by mass-production injection molding. Versatility and capability are supported by providing pressure support modes and the timing and pressure ranges needed to support a wide range of people suffering from different forms of respiratory distress. These requirements resulted in a hydro-pneumatic ventilator capable of use with patients experiencing ARDS, without any electronics.

A priori risk analysis identified a wide range of risks to the patient but focused on reducing the likelihood of adverse events caused by the BubbleVent. Adverse event risks appeared most likely to arise from a failure to provided ventilation and the unintended and prolonged delivery of high pressures to patients’ lungs. The risk of infection to and from the patient was also identified as a key risk.

The BubbleVent was designed in SOLIDWORKS 2020/2021 (Dassault Systèmes, Vélizy-Villacoublay, France). Parts were exported from SOLIDWORKS in STL or AMF file formats and were converted into GCODE toolpaths for printing using Simplify3D v4.0 and v4.1 (Simplify3D, Cincinnati, OH, USA). For printing, OctoPrint (v1.3.6, Gina Häußge et al.) was used to control the MakerGear M3 ID 3D FDM printer (MakerGear, Beachwood, OH, USA).

The BubbleVent breathing circuit valve (Figure 2) can be manufactured using a single nozzle, material extrusion/fused deposition (FDM) 3D printer. FDM, also known as material extrusion, is a process used to make thermoplastic parts through heated extrusion and deposition of materials in a series of layers. The part is fabricated layer-on-layer, with the high temperature of extruded thermoplastic enabling the layers to bond to one another. The part is complete when all the layers are extruded and the part has cooled. Vat polymerization (Form3B, Formlabs, Somerville, MA, USA) and MultiJet Fusion (Hewlett Packard, Palo Alto, CA, USA) processes were also used to manufacture test parts for the BubbleVent. These test parts were not included in the BubbleVent described in this publication but are considered suitable manufacturing processes. These processes both created test parts that were durable, airtight, and within manufacturing tolerances.

Other parts such as the PIP, PEEP, and BCV tanks and tank lids can be made from a range of commercially available materials, including cast clear acrylic tubes (Tap Plastics, San Leandro, CA, USA). Some commonly available commercial components, such as 1” vinyl tubing and tubing connectors (Home Depot, Atlanta, GA, USA) are needed to fabricate a complete device. The nature of 3D printing and the BubbleVent design also means unavailable commercial components can be printed. The interfaces of the BubbleVent can also be easily modified to utilize commercial components that are available.

The BubbleVent is designed to be produced using mass-manufacture techniques and consultation with multiple injection molding experts confirmed the BubbleVent could be molded from common materials such as polypropylene with minimal modifications to the current design.

#### 2.2.1. Valve System

The BubbleVent was printed using a single, uncalibrated MakerGear M3-ID fused deposition modeling (FDM) printer. The print material was 1.75 mm polylactic acid (PLA) filament (MakerGear). PLA was also used to print support material when required. Whenever possible, when preparing toolpaths in Simplify 3D, the default print MakerGear PLA process profile settings (M3-ID Rev1 PLA 1.5 process profile) were used. A layer height of 0.02 mm was used for all parts, with an extrusion temperature set at 222 °C, with a range of 215–225 °C. Extrusion rate varied from 2600–3000 mm/min. Smaller parts used lower print speeds, primarily to minimize the risk of the part being pulled from the print bed during printing. When necessary, default process settings were adjusted within a narrow range based on part size, function, geometry, orientation on the build plate, and the number being printed in a single build. Settings that were varied were infill percentage (15–35%), number of solid top and bottom layers (2–8), the use of a skirt/brim (yes/no), and use of support structures (yes/no).

Printed parts were manually inspected using digital Vernier calipers and, if necessary, post processed and modified using hand tools and abrasive paper. Sliding surfaces were smoothed using files and abrasive paper (80–220 girt) to reduce layer lines resulting from the FDM printing process. Valve surfaces were also smoothed using custom, 3D-printed valve surfacing tools.

For the BubbleVent tested by the authors, float gas flow tubes were made from both printed and 1/2” Apollo PEX pipe (Home Depot, Atlanta, GA, USA). PEX pipe provided a smooth surface on which the float air chamber would slide with minimal postprocessing. These pipes were joined to printed parts using consumer-grade adhesive to deliver the desired part design.

Between 50 g and 120 g of weight was added to the float air chamber to help the float sink. Weight was added in the form of galvanized or stainless-steel washers, nuts, and/or heavy gage copper wire. Copper wire became the preferred weighting option due to its availability, corrosion resistance, cost per kilo, and recyclability. The PEEP and PIP timing control linkages were weighted using two 1/4” SAE stainless steel nuts to ensue these linkages latched consistently. The “float within a float”, found inside the float air chamber, was weighted using a 5/8” SAE washer

Approximately 0.40 kg of PLA filament was used to produce the breathing circuit valve, including any support structure material required. At USD 30/kg of PLA filament, the BubbleVent costs USD 12 in materials. While the total print time and the time spent postprocessing parts were not formally measured, the authors estimated that it takes 43 h to print and 2 h to post process the breathing circuit valve parts, respectively. The 43 h estimate is based on 1 kg PLA = 300 m, 0.40 kg = 120 m, and 120 m@2800 mm/min = 42 + 1 h total for manual print setup and part removal. Including a 20% variance in estimate and actual print times, the entire process of building the device requires 57 h. At USD $15/h, print setup, and post processing are expected to be the key costs of manufacturing the BubbleVent using FDM. In contrast, consultation with multiple injection molding experts indicated a potential cost of goods of USD $30, with 15 min of assembly time.

#### 2.2.2. Water Tanks

The housing the BubbleVent tested by the authors the PIP/PEEP/valve system tanks are made from two 5” and one 8” diameter and acrylic cylinders (Tap Plastics, Seattle, WA, USA), respectively. The cylinders are capped with an acrylic disc on the ends. Custom brackets were 3D printed using FMD and mechanically position and link the tanks. Tank lids were also 3D printed using FDM. All FDM printing followed the processes used for printing the breathing circuit valve parts.

The 1 × 1/8” PVC tubing and matching T- and L-connectors (Home Depot) were used to create the breathing gas pathway to and from the breathing circuit valve. The 1 × 1/8” PVC tubing was also used to create the PIP and PEEP tubes. Other supplies, such as 1/4” OD/.170” ID clear vinyl tube (Home Depot, Atlanta, GA, USA) were used in the BubbleVent to enable easy control of the water level.

Diffusers were designed and printed using the FDM process used for the breathing circuit valve parts. These diffusers were added to the PIP and PEEP tubes to moderate the large pressure oscillations created by gas exiting the PIP and PEEP tubes at high rates (20–80 LPM).

### 2.3. In Vitro Testing of BubbleVent

#### 2.3.1. Experimental Setup

We designed and conducted studies in vitro in three separate tests, the first two assessing the capability of BubbleVent and the third comparing BubbleVent to a critical care ventilator commonly used in North American hospitals. Adult patients were modeled using the ASL 5000 Test Lung (Ingmar Medical, Pittsburgh, PA, USA), a digitally controlled, high-fidelity breathing simulator, which utilizes a screw-drive controlled piston and mathematical modeling to simulate size and disease specific pulmonary mechanics (Figure 2). Table 2 lists the lung model parameters used for three different lungs models aimed at various stages of ARDS. The pulmonary mechanical values were based on those from Arnal et al. and modified to values that we felt were indicative of the range of ARDS severity in COVID-19 patients [27]. The T_I_ and RR values in Table 2 were modified from Ntoumenopoulus et al. as values indicative of stable mechanical ventilation [28]. The lung model was connected to the ventilator using a 7.5 mm diameter endotracheal tube (DYNJAAUC75, Medline Industries, Northfield, IL, USA), which was connected to the ventilator and gas supply using an adult airway circuit (RT110, Fisher & Paykel, Auckland, New Zealand). No humidifier was used for this study.

#### 2.3.2. Test 1: Timing Control

Throughout the test, PIP was set to 25 cmH_2_O, and PEEP was set to 10 cmH_2_O. The lung model compliance, inspiratory resistance, and expiratory resistance were set to the normal model parameters given in Table 2. Test 1 was divided into two parts. For Part 1, BCV exhaust resistance and float flow were varied while measuring T_I_, T_E_, and RR at the lung model. Resistance was increased by plugging the end of the float exhaust with 3D-printed caps that possessed orifices with decreasing cross-sectional areas. Five resistors were tested using cross-sections ranging from 71.3 mm^2^ to 2.0 mm^2^ in area. Five float flows were tested and varied independently from 1.5 LPM to 20 LPM. T_I_, T_E_, and RR were computed by the ASL 5000 and recorded. Each setting of FER and float flow were tested for a total of 25 runs. The sampled data for each run of 10 breaths (*n* = 10) was extracted from the ASL 5000 software and saved to a spreadsheet (Excel, Microsoft, Redmond, WA, USA). Mean and standard deviation were calculated for 10 simulated breaths using MATLAB (Mathworks, Natick, MA, USA).

For Part 2, the BCV exhaust resistance was varied using a ball valve that was adjusted manually to change the size of the valve opening. Using the same setup, the accuracy and precision of the monitoring system in computing T_I_ and RR was compared with the values recorded by the ASL 5000. Nineteen T_I_ values were tested ranging from 0.5 s to 4.0 s and RR values were tested ranging from 5 BPM to 50 BPM. The output of the BVMS was noted and recorded 5 different times (*n* = 5) during the ten-breath epoch of simulated breaths. The mean T_I_ and RR values of the BVMS system were calculated and compared with the measured T_I_ and RR at the ASL 5000 using a percent error computation.

#### 2.3.3. Pressure Control and Volume Delivery

Normal lung model mechanics in Table 2. The BubbleVent was tested throughout a range of PIP and PEEP pressures. I:E ratio was kept consistent throughout the test at 1:2 with T_I_ set at 1.0 s and T_E_ set at 2.0 s for a RR of 20 BPM. T_I_ and RR were tuned based on real-time output computed by the ASL 5000 to achieve as close as possible to the intended I:E and RR. PEEP and PIP were initially set to 5 cmH_2_O and 20 cmH_2_O, respectively, by visually inspecting the height of the water level in each tank, and bias flow was set to 40 LPM as indicated by Sequoia. The model was allotted 30 s to stabilize after the adjusting parameters. The bias flow was adjusted sequentially to 60 LPM and 80 LPM for two more recordings. PIP was increased to 25 cmH_2_O, and the same three bias flows were tested; this was repeated at PIPs of 30 and 35 cmH_2_O. The same range of bias flows and PIPs were repeated at a PEEP of 10 and 15 cmH_2_O. Altogether, 36 runs were analyzed. The sampled data for each run of 10 breaths (*n* = 10) was extracted from the ASL 5000 software and saved to a spreadsheet (Excel, Microsoft, Redmond, WA, USA). Mean and standard deviation were calculated for 10 simulated breaths using MATLAB (Mathworks, Natick, MA, USA). Each ten-breath run in the final dataset had the mean and SD of the T_I_, RR, PIP, PEEP, and V_T_ computed. PIP and PEEP were compared using percent error from the intended setting (hydrostatic depth of the tube).

#### 2.3.4. Comparison with a Critical Care Ventilator

The BubbleVent was compared with a hospital-standard critical care ventilator (VN500, Draeger Medical, Lubek, Germany). The BubbleVent performance is bias flow dependent, so the BubbleVent testing was distinguished by three different bias flow rates of 40, 60, and 80 LPM. As each rate provided different pressure and volume delivery characteristics, for the purposes of testing the BubbleVent, they were assessed in BV40, BV60, and BV80 modes, respectively, with mode corresponding to bias flow rate. These three modes were compared with the VN500. The modes were tested at each lung model outline in Table 2, representing patients with increasing severity of ARDS (Normal, Mild, and Severe ARDS). The sampled data for each run of 10 breaths (*n* = 10) was extracted from the ASL5000 software and saved to a spreadsheet (Excel, Microsoft, Redmond, WA, USA). Mean and standard deviation were calculated for 10 simulated breaths using MATLAB (Mathworks, Natick, MA, USA). Normality for each run was determined by a Kolmogorov–Smirnov test. Differences were tested using ANOVA followed by Student–Neumann–Keuls post hoc analysis. Statistical significance was set at *p* < 0.05.

## 3. Results

### 3.1. Test 1 Results

Part one of Test 1 results are given in Figure 3. A nonlinear relationship was observed between orifice cross-section and T_I_. As orifice cross-section increased from 2.0 mm^2^ to 71.3 mm^2^, the T_I_ decreased sharply from 4.69 s to 0.4 s with a float flow of 1.5 LPM. The range of T_I_ at 20 LPM float flow was from 2.35 s to 0.46 s, indicating an interaction between float exhaust resistance and float flow on T_I_ and T_E_. T_E_ did not change with changes in increase from 1.8 s to 2.1 s as cross-section increased from 10 mm^2^ to 20 mm^2^ and remained at 2.1 s as FER increased.

Part two of Test 1 results are given in Table 2. The BVMS T_I_ values consistently overshot the measured T_I_ values recorded by the ASL 5000. The percent error between the set and measured T_I_ ranged from −1.6% to 35.3%. Only two T_I_ settings were within the 5% acceptable error rate. The BVMS RR values were much closer to the measured RR values. Percent error values ranged from −5.7% to 8.7%, and all but two settings fell outside the 5% acceptance criteria. Wave tracings are shown in Figure 3, which illustrates how the timing control of the float exhaust resistance and float flow result in different I:E ratios and RRs.

### 3.2. Test 2 Results

Test 2 results are given in Table 3. Throughout the entire parameter set, T_I_ varied in the range of 1.26 s to 1.39 s and RR between 14.5 BPM to 16.0 BPM. Table 4 provides the Mean ± SD and percent error of PIP and PEEP, each ten-breath sample for every combination of PIP, PEEP, and bias flow. The percent error between the set and measured PIP ranged from −19.19% to 57.44%. The percent error of the PEEP ranged from 10.50% to 145.56%.

Figure 4 shows the V_T_ delivered at each PIP and PEEP pressures measured in the lung model with respect to the set PIP, PEEP, and bias flow values. Tidal volumes ranged from 120 mL when PIP was set to 20 cmH_2_O and PEEP to 15 cmH_2_O, to 950 mL at PIP of 35 cmH_2_O and a PEEP of 5 cmH_2_O. The largest tidal volumes were observed at bias flow rates of 60 LPM with PIP pressures at 35 cmH_2_O, while the lowest tidal volume recorded was at 80 LPM with a PIP of 20 cmH_2_O.

### 3.3. Test 3 Results

Figure 5 provides a small sample of the BubbleVent and VN500 P_AW_, inspiratory flow, and V_T_ tracings for visual comparison. Oscillations were evident in the BubbleVent tracings as opposed to lack of noise in the Draeger VN500.

Figure 6 shows the outcomes for the comparison among the different ventilators at increasing severity of ARDS. The Draeger VN500 was significantly different than BV40, BV60, and BV80 in all outcomes (*p* < 0.01). The Draeger VN500 provided higher volumes on lower PIP and PEEP pressures (*p* < 0.005). BV40, BV60, and BV80 were not different in RR for all lung models (*p* > 0.05). BV40 and BV60 provided a lower PIP than BV80 for the normal model and severe ARDS model (*p* < 0.05). BV40 provided a lower PIP than BV60 and BV80 for the mild ARDS model. BV40 and BV60 provided a lower PEEP and higher V_T_ than BV80 for all lung models (*p* > 0.05). BV40 and BV60 were similar in delivery of PEEP and V_T_.

## 4. Discussion

The major finding of this study is that the BubbleVent offers a level of control over the types and range of breaths that does not appear to have been offered by hydro-pneumatic respiratory support technology. Test 1, the T_I,_ and RR outcomes from the BubbleVent were extensive, offering T_I_ from 0.4 s to 4.7 s and RR from 5 BPM to 50 BPM. While the timing control offered by the BubbleVent provided a novel way of breath delivery without the need for microprocessor control or electronic valves, the delivered PIP and PEEP pressures were not reliable at providing the pressures intended by the user. In Test 2, PIP and PEEP outcomes were not within the 5% criteria for all the intended settings. In Test 3, the Draeger VN500 outperformed the BubbleVent in all outcomes. Despite its performance relative to a standard critical care ventilator, as a prototype, 3D-printed, hydro-pneumatic device, the BubbleVent can provide sufficient ventilation to support a simulated patient suffering from ARDS.

The outcomes of the timing test were promising, demonstrating the ability for a pneumatically powered valve to generate sophisticated breathing waveforms. While the BCV does not appear to completely decouple to the T_I_ and T_E_, the decoupling appears sufficient for clinically relevant control of T_I_ and RR for PIP. The variance of the T_I_ and RR were very small as well, indicating that the precision of the timing mechanism is suitable for clinical use. However, in comparison with the precision of the Draeger VN500, the BubbleVent shows a significant difference in T_I_ and RR. While the difference in precision is notable, the questions arises whether the timing precision of breaths is good for long term outcomes. Ventilator-induced lung injury (VILI) has been noted by others because of consistent breaths not accounting for the variable breathing patterns of the patient [29]. The BubbleVent could potentially offer the precision to match the patient as well as the flexibility for changing breathing patterns. The hydropneumatics design of the BubbleVent also prevents large pressures being transmitted to the patient quickly and unintentionally.

The Draeger VN500 provided significantly larger volumes, lower PIP pressures, and lower PEEP pressures than the BubbleVent. The Drager VN500 volumes and pressures are also closer to the intended device settings than the BubbleVent. It is evident that the microprocessor-based design of the VN500 offers a level of control that an analog, low-cost ventilator may struggle to compete with.

However, limitations of the BubbleVent design can also account for the discrepancy in delivered pressures and volumes. For pressure delivery, the large bias flow rates create back pressures. The differences in pressure are largely flow dependent, a consideration that is routinely accounted for in B-CPAP use in infants [18,19]. Large flow rates generate back pressures that need to be accounted for in the depth setting of the PIP and PEEP tubes. For example, a bias flow of 50 LPM requires setting the depths to less than the intended PIP and PEEP pressures. A scale relating bias flow to PIP and PEEP depth setting could be developed by titrating the flow and measuring PIP and PEEP pressure. With the accompanying scale, the user can mitigate delivering increased PIP and PEEP pressures when setting the BubbleVent to the patient. See the “ISO 80601 Pressure Ventilation Testing” section of the Appendix A for performance testing where the water level was titrated to the desired pressure level.

Discrepancies in volume delivery between the Draeger VN500 and BubbleVent are a likely result of two phenomena. First is the volume loss in pressurizing the BubbleVent when the BCV closes. After the BCV closes, the PIP tube needs to be filled and bubbling before the PIP pressure is achieved. Pressurization includes filling the volume of the patient circuit, BCV chamber, and PIP tube. In comparison, the VN500’s microprocessor accounts for the patient circuit volume when ventilating. Furthermore, the minute capacitance of the steel valves relative to the tubing used in the BubbleVent results in more volume delivery to the patient. The second phenomenon is how airtight the component interfaces are in the BubbleVent. Leaky interfaces, along with the increased pressures, result in reduced delivered V_T_ to the patient (Figure 6b,c). Equipping the BubbleVent with more rigid and airtight components can help alleviate volume losses more effectively.

Lastly, while volume delivery per breath from the BubbleVent was less than the Draeger VN500, the superimposed oscillations provided by the BIMV may offer clinical benefits not available to patients on a valve-based ventilator. DiBlasi et al. observed lung recruitment and gas exchange in an infant animal model of infant ARDS when providing large oscillations with high amplitude bubble CPAP, pressure oscillations close in size to those delivered by the BubbleVent [19]. Despite the differences in lung volume between adults in infants, at the alveolar level there may be small incidents of recruitment. Prolonged exposure to BIMV with the BubbleVent could lead to notable lung recruitment in adults.

### 4.1. Design Risks and Limitations

The risk to patients from ventilator failure, especially during continuous mandatory ventilation, requires a very high degree of reliability and accuracy from a respiratory support device. A ventilator must work as intended and expected every time it delivers a breath. Device failure could lead to a patient experiencing a prolonged exposure to PIP pressures. While rare, these failures need to have mitigations put in place to reduce possible harm to a patient. Additional improvements to device reliability and accuracy are warranted based on the test results shown above.

Durability of the breathing circuit valve and the impact of fatigue and friction on bearing surfaces remains largely undetermined. While the authors estimate in testing the BubbleVent for this publication is based on a usage of 10 × 8 h days at varying breaths per minute (BPM) rates, the final BubbleVent design is expected run continuously for one million cycles (e.g., 24 days of continuous use at 30 breaths per minute). The impact and challenges of running the BubbleVent in a clinical setting has also not been tested.

Ventilators are expected to have several alarm systems to alert the clinical team to variances from the intended device settings. The design of the BubbleVent means a clinician can visually inspect the device to verify pressure settings and timing, but this may present a challenge to care teams accustomed to digital displays and automated alerts and alarms. The authors noted the BubbleVent could benefit from auxiliary pressure monitors and alarm systems that are indicated for used with B-CPAP systems.

The BubbleVent offers effective and easy-to-manage pressure targeting; the depth of the tube in the water determines the pressure delivered to the patient and is clearly visible. However, volume targeting modes are more commonly used when ventilating adults. Delivered volumes can be calculated from the inspiratory time and flow rate, but accurate, manual calculations require time and effort from the care team. A nomogram or other digital calculator could be offered for use when the BubbleVent is being used in a volume-targeting mode.

The BubbleVent is not entirely self-contained and requires an external air source. The authors tested various air sources with various version of the BubbleVent, including using consumer air compressors, centrifugal air blowers, and membrane pond pumps. All methods demonstrated success, with the pond pump offering a balance of suitable air flow rates (10–80 LPM) and low noise generation. Additional work is required to determine which air sources are most suitable for use with the BubbleVent.

### 4.2. Study Limitations

This study was carried out in vitro and does not have physiologic data to corroborate the performance of the BubbleVent in a live patient with ARDS. Furthermore, the BubbleVent was tested only at a few different C_L_, R_I_, and R_E_ settings, more intermediate ARDS settings would be worth using to encompass a larger patient population. All lung models were passive and did not have spontaneous breathing patterns. Bench-top studies such as this demonstrate efficacy of a device in ideal conditions, but in low-resource settings where staff and biomedical engineering support are limited, it is difficult to create ideal conditions. Additional studies are needed to see how effective the system is when integrated into healthcare facilities in different contexts in LMICs.

## 5. Conclusions

The BubbleVent is a novel ventilator that uses B-CPAP and Bubble-BiPAP principles to deliver PSV in patients in critical care settings. While further testing is required to verify the effectiveness of the BubbleVent, this report provides proof-of-principle and conceptual evidence supporting the feasibility of implementing bubble-based airway support at positive airway pressures suitable for adult continuous mandatory ventilation. On the engineering side, the BubbleVent attests to what is possible with additive manufacturing, exhibiting the ability for prospective medical devices to be printed on demand.

## Figures and Tables

**Figure 1 bioengineering-09-00151-f001:**
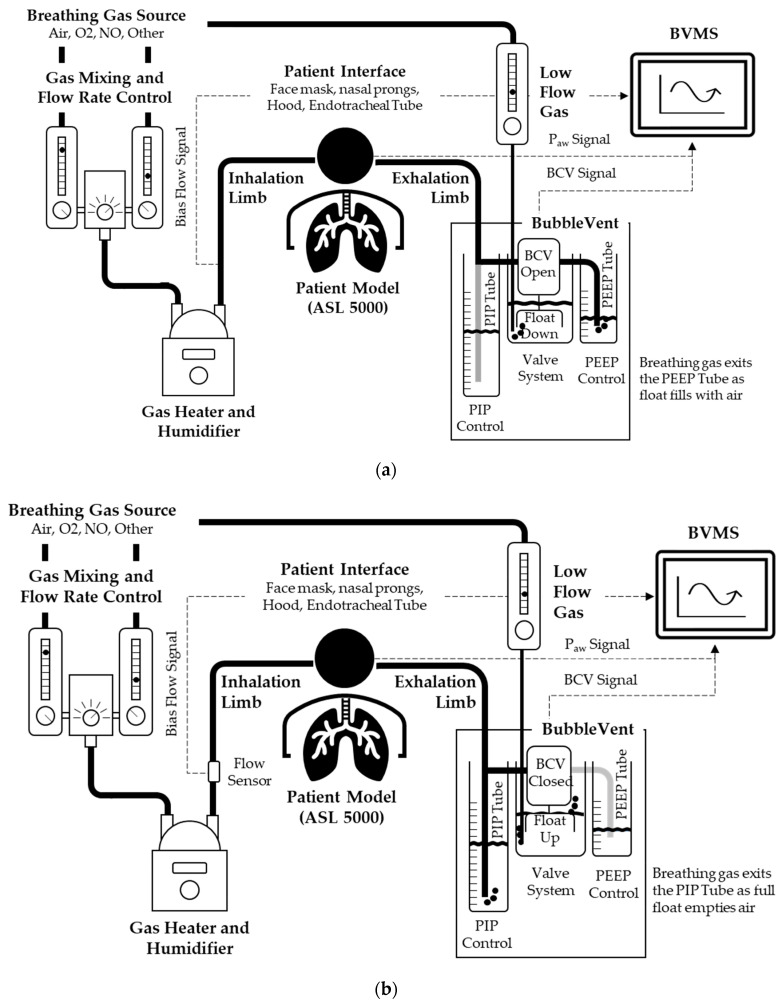
A schematic of the BubbleVent system including the directional characteristics of gas flow during inhalation and exhalation phases. (**a**) Exhalation Phase/B-CPAP mode: PEEP is delivered to the patient model; the BCV mechanism is open; gas is entering the float. (**b**) Inhalation phase: PIP is delivered to the patient model; the BCV is closed; gas is exiting the float.

**Figure 2 bioengineering-09-00151-f002:**
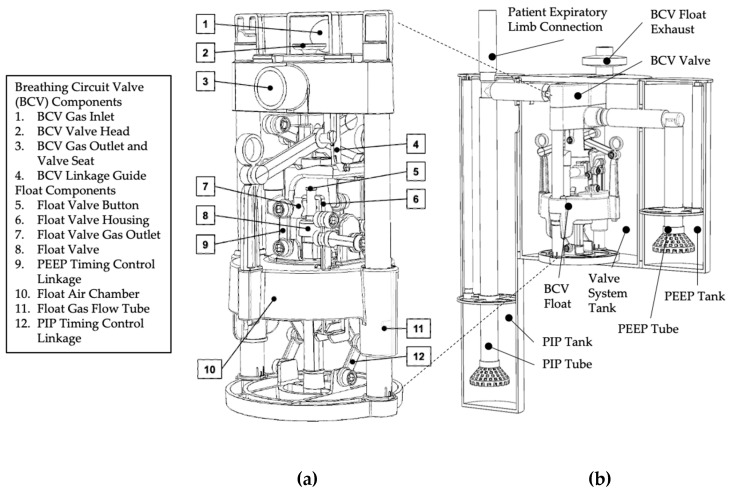
A diagram of the valve system and complete BubbleVent device. (**a**) *Breathing Circuit Valve Detail*: the key sub-assemblies and parts of the valve system; the float components are submerged below the waterline. (**b**) *BubbleVent Detail*: The key assemblies within the BubbleVent device; PIP and PEEP tube ends are submerged below a controlled water level in the PIP and PEEP tanks.

**Figure 3 bioengineering-09-00151-f003:**
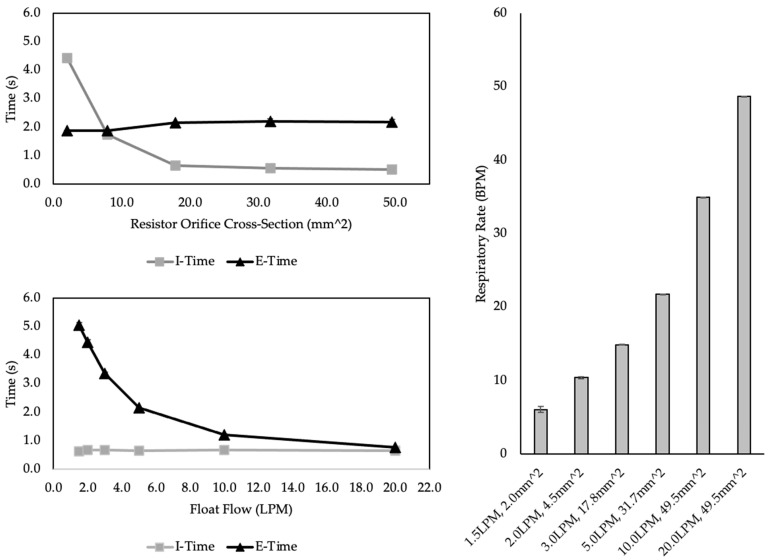
Outcomes of Part 1 of Test 1. The top left graph shows the trend of increasing float exhaust resistance on T_I_ and T_E_. The bottom left graph shows the relationship between float flow and T_I_ and T_E_. The graph to the right demonstrates how RR changes with changes in float exhaust resistance and float flow. All data points and bars are mean ± SD of 10 breaths. Lung model compliance was 55 mL/cmH_2_O, inspiratory and expiratory resistance was 12 cmH_2_O/L/min.

**Figure 4 bioengineering-09-00151-f004:**
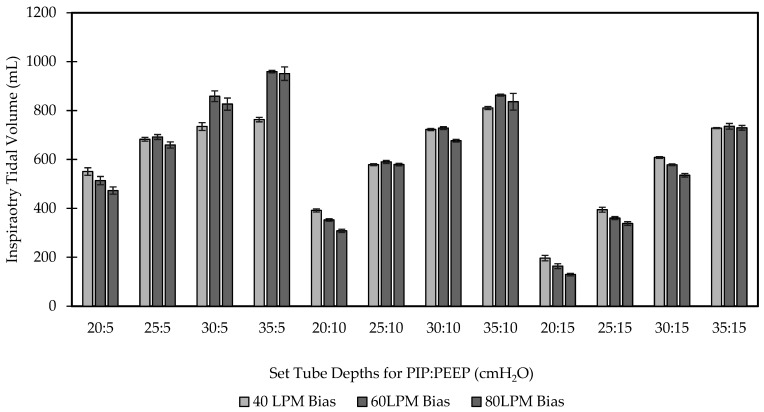
Delivered volumes to the lung model in response to changes in bias flow, PIP, and PEEP. PIP and PEEP are indicated by the numbers just below the horizontal axis. PIP and PEEP are separated by a “:” as, for example, 20:5 is PIP of 20 cmH_2_O and a PEEP of 5 cmH_2_O. Lung model compliance was 55 mL/cmH_2_O, and inspiratory and expiratory resistance was 12 cmH_2_O/L/min. All bars are Mean ± SD.

**Figure 5 bioengineering-09-00151-f005:**
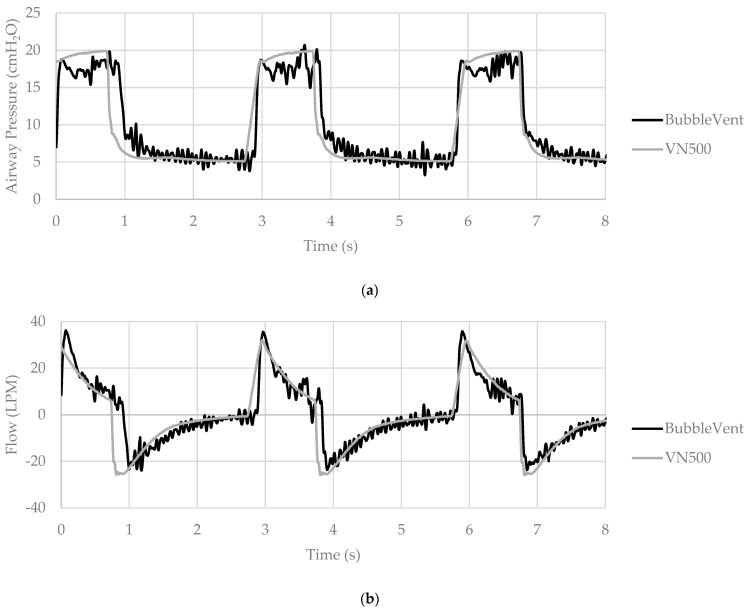
Waveform tracings of the BubbleVent and Draeger VN500 for three breaths. Lung model compliance and resistance was set to 20 mL/hPa and 20 hPa/L/s, PIP to 20 cmH_2_O, and PEEP to 5 cmH_2_O; RR = 20 BPM and T_I_ = 1.0 s. Figure (**a**) is airway pressure, (**b**) is inspiratory flow, and (**c**) is volume.

**Figure 6 bioengineering-09-00151-f006:**
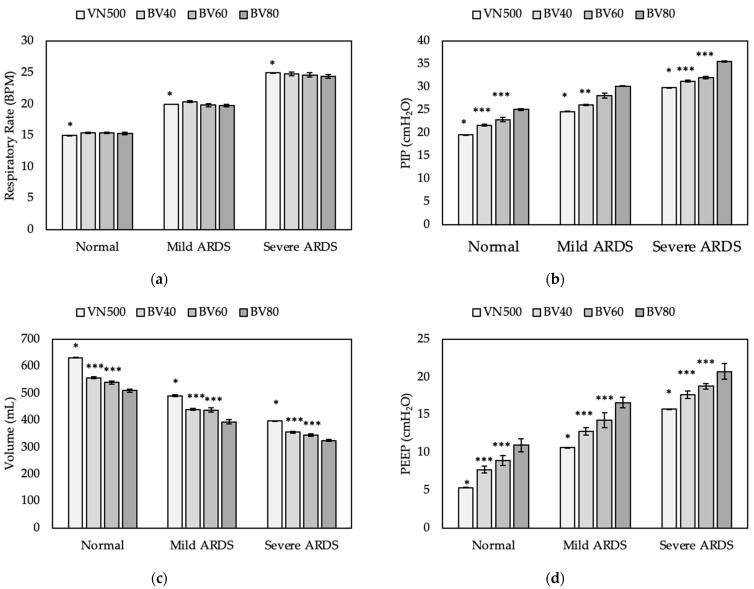
Outcomes of comparison test between the BubbleVent at different bias flow rates and Draeger VN500. Graph (**a**) compares the RR of the 4 ventilators, graph (**b**) compares the measured PIP outcomes, graph (**c**) compares the volume delivery outcomes, and graph (**d**) compares the measured PEEP values. A “*” indicates significantly different than all three ventilators; a “**” indicates significantly different than BV60 and BV80; a “***” indicates significantly different than BV80 only. All bars are mean ± SD.

**Table 1 bioengineering-09-00151-t001:** The relationship between the pressure delivered and the states of key bubble vent Sub-Assemblies and Parts.

Pressure State	BCV State	Valve Head State	Float State
PIP	Closed	Down	Up
PEEP	Open	Up	Down

**Table 2 bioengineering-09-00151-t002:** Lung model parameters and ventilator settings for simulating ARDS in adult patients [17,18].

**Parameter**	**No ARDS**	**Mild ARDS**	**Severe ARDS**
Lung Compliance (mL/cmH_2_O)	55	45	35
Inspiratory Resistance (cmH_2_O/L/min)	12	12	12
Expiratory Resistance (cmH_2_O/L/min)	12	13	14
**Setting**	**No ARDS**	**Mild ARDS**	**Severe ARDS**
Set PEEP (cmH_2_O)	5	10	15
Set PIP (cmH_2_O)	20	25	30
Set T_I_ (s)	1.3	1.0	0.8
Set RR (BPM)	15	20	25

ARDS = acute respiratory distress syndrome; PEEP = positive end expiratory pressure; PIP = peak inspiratory pressure; T_I_ = inspiratory time; RR = respiratory rate.

**Table 3 bioengineering-09-00151-t003:** Outcomes of Part 2 of Test 1. Measured T_I_ and RR of the patient at different settings of T_I_ and RR. T_I_ and RR were set using the BubbleVent monitoring system (BVMS) and measured using the ASL 5000.

Set T_I_ (s)	Set RR (BPM)	BVMS T_I_ (s)	Measured T_I_ (s)	T_I_ Percent Error (%)	BVMS RR (BPM)	Measured RR (BPM)	RR Percent Error (%)
0.5	15.0	0.56 ± 0.01	0.46 ± 0.05	22.0	14.99 ± 0.08	14.9 ± 0.08	0.6
0.5	20.0	0.55 ± 0.05	0.56 ± 0.03	−1.6	19.93 ± 0.33	19.52 ± 0.05	2.1
0.5	30.0	0.53 ± 0.04	0.5 ± 0.01	5.4	30.51 ± 0.39	30.66 ± 0.03	−0.5
0.5	40.0	0.53 ± 0.04	0.48 ± 0.01	11.2	40.38 ± 1.26	40.13 ± 0.02	0.6
0.5	50.0	0.58 ± 0.01	0.43 ± 0.01	35.3	50.27 ± 0.64	50.28 ± 0.01	0.0
0.8	8.0	0.79 ± 0.03	0.73 ± 0.08	8.8	8.14 ± 0.23	7.88 ± 0.25	3.3
0.8	16.0	0.79 ± 0.01	0.66 ± 0.01	20.6	16.08 ± 0.18	16.27 ± 0.04	−1.2
0.8	24.0	0.84 ± 0.04	0.79 ± 0.02	6.0	24.01 ± 0.48	24.19 ± 0.02	−0.7
1.0	5.0	0.99 ± 0.02	0.88 ± 0.03	12.0	5.26 ± 0.23	5.3 ± 0.12	−0.9
1.0	10.0	0.99 ± 0.01	0.86 ± 0.03	15.9	10.06 ± 0.14	10.16 ± 0.05	−1.0
1.0	15.0	0.96 ± 0.05	0.88 ± 0.01	9.2	14.99 ± 0.25	14.84 ± 0.04	1.0
1.0	20.0	1.02 ± 0.07	0.9 ± 0.05	13.4	19.78 ± 0.33	19.5 ± 0.08	1.5
1.0	30.0	1.08 ± 0.02	0.92 ± 0.02	17.4	30.55 ± 0.65	30.74 ± 0.03	−0.6
1.5	10.0	1.55 ± 0.08	1.39 ± 0.09	11.6	9.43 ± 0.65	10.01 ± 0.15	−5.7
1.5	20.0	1.59 ± 0.03	1.49 ± 0.04	6.6	21.01 ± 0.17	20.96 ± 0.04	0.2
1.5	30.0	1.46 ± 0.04	1.2 ± 0.02	21.0	30.3 ± 0.66	29.97 ± 0.02	1.1
2.0	20.0	1.97 ± 0.04	1.76 ± 0.05	11.8	20.63 ± 0.35	20.58 ± 0.05	0.2
3.0	15.0	2.97 ± 0.04	2.88 ± 0.15	3.3	14.77 ± 0.18	14.6 ± 0.15	1.2
4.0	10.0	3.92 ± 0.15	4.2 ± 0.34	−6.7	10.21 ± 0.31	9.39 ± 0.42	8.7

BVMS = BubbleVent Monitoring System; T_I_ = inspiratory time; RR = respiratory rate.

**Table 4 bioengineering-09-00151-t004:** Outcomes of Test 2. Measured PIP and PEEP when settings are PIP, PEEP, and bias flow settings are changed. Measured PIP and PEEP are given in mean ± SD and percent error is given as the deviation from the set value.

Set PEEP (cmH_2_O)	Set PIP (cmH_2_O)	Bias Flow (LPM)	Measured PIP (cmH_2_O)	PIP Percent Error (%)	Measured PEEP (cmH_2_O)	PEEP Percent Error (%)
5	20	40	21.59 ± 0.25	7.94	7.96 ± 0.30	59.15
5	20	60	23.37 ± 0.26	16.83	9.90 ± 0.88	97.93
5	20	80	26.43 ± 0.25	32.17	12.39 ± 0.77	147.78
5	25	40	24.93 ± 0.16	−0.28	7.82 ± 0.63	56.32
5	25	60	27.32 ± 0.39	9.27	9.26 ± 0.53	85.16
5	25	80	29.23 ± 0.56	16.90	11.61 ± 0.9	132.25
5	30	40	27.97 ± 0.12	−6.77	8.12 ± 0.53	62.50
5	30	60	30.67 ± 0.47	2.24	9.52 ± 0.78	90.38
5	30	80	32.64 ± 0.51	8.80	12.28 ± 0.99	145.56
5	35	40	28.28 ± 0.12	−19.19	7.72 ± 0.55	54.30
5	35	60	34.09 ± 0.08	−2.61	9.46 ± 0.77	89.28
5	35	80	35.65 ± 0.54	1.84	11.95 ± 0.67	139.06
10	20	40	22.74 ± 0.30	13.71	12.66 ± 0.41	26.57
10	20	60	25.91 ± 0.12	29.53	14.43 ± 1.02	44.33
10	20	80	30.10 ± 0.13	50.48	16.92 ± 0.88	69.20
10	25	40	26.93 ± 0.43	7.74	12.96 ± 0.61	29.64
10	25	60	28.56 ± 0.49	14.23	14.33 ± 0.82	43.31
10	25	80	26.84 ± 0.42	7.37	12.75 ± 0.25	27.54
10	30	40	30.23 ± 0.13	0.76	12.95 ± 0.43	29.48
10	30	60	32.61 ± 0.35	8.69	14.45 ± 0.68	44.47
10	30	80	33.92 ± 0.51	13.06	16.84 ± 0.43	68.44
10	35	40	34.03 ± 0.03	−2.78	12.84 ± 0.55	28.38
10	35	60	35.75 ± 0.31	2.14	15.07 ± 0.28	50.74
10	35	80	36.62 ± 0.78	4.63	16.58 ± 1.15	65.78
15	20	40	24.62 ± 0.20	23.10	17.71 ± 0.53	18.06
15	20	60	28.77 ± 0.44	43.83	19.41 ± 1.17	29.43
15	20	80	31.49 ± 0.84	57.44	20.67 ± 0.83	37.80
15	25	40	27.35 ± 0.28	9.38	17.20 ± 0.52	14.67
15	25	60	30.40 ± 0.16	21.61	18.92 ± 0.71	26.10
15	25	80	34.33 ± 0.26	37.30	19.98 ± 0.58	33.23
15	30	40	31.02 ± 0.09	3.40	16.57 ± 0.46	10.50
15	30	60	33.32 ± 0.61	11.08	18.77 ± 0.89	25.16
15	30	80	34.45 ± 0.38	14.83	20.95 ± 0.52	39.68
15	35	40	35.22 ± 0.10	0.63	17.45 ± 0.63	16.33
15	35	60	37.23 ± 0.64	6.38	19.36 ± 0.71	29.07
15	35	80	38.90 ± 0.67	11.16	20.91 ± 0.96	39.43

PEEP = positive end expiratory pressure; PIP = peak inspiratory pressure.

## Data Availability

Please reach out to the corresponding author for data availability.

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
