# Peer review of "Performance Characteristics of a Novel 3D-Printed Bubble Intermittent Mandatory Ventilator (B-IMV) for Adult Pulmonary Support"

_bioengineering, 2022, doi:10.3390/bioengineering9040151_

Round 1

Reviewer 1 Report

The authors have modified the text of the manuscript according to the suggestions of the reviewer, significantly improving the quality of the presentation and of the technical details.

Author Response

We cannot thank the reviewer enough for their time and suggestions. Your comments improved the manuscript substantially.

Reviewer 2 Report

Suggest the authors to provide a brief description of the FDM process. 

Author Response

Thank you for your review of our manuscript. We really appreciated your suggestion and have added a few more statements about the FDM printing process in Section 2.2 Paragraph 4. 

This manuscript is a resubmission of an earlier submission. The following is a list of the peer review reports and author responses from that submission.

Round 1

Reviewer 1 Report

  • Authors uses 3 D printing technology but in the entire article noting significant contributions made towards it.

Authors write:  

 3D printed Bubble Inter-mittent Mandatory Ventilator (B-IMV) and about the apparatus currently composed of 3D printed expiratory valve components combined with commonly available tubes, hoses, and tubing connectors to create an air-powered, hydro-pneumatic, PPV generator.

Authors fails to provide any substernal evidence in support of the above  

  • The Authors:  Hypothesized that the TI, RR, PIP, PEEP, and VT delivered to a mechanical model configured with normal lung mechanics and mild-severe ARDS disease are not different between Bubble Vent and a critical care ventilator.

The basis of the made hypothesis to be supported by the facts.

  • CFD technique and the use of the commercial packages to be used to evaluate the different parameters to support the results obtained. It may give the adequate information about the flow.
  • Conclusions drawn are very superficial. It need to be re-drafted

Reviewer 2 Report

Manuscript ID: bioengineering-1583213

As stated by the authors the purpose of this paper is to: “describe the iterative design and performance characteristics of the BubbleVent, a novel 3D printed valve system combined with commonly found tubes, hoses, and connectors, can provide intermittent mandatory ventilation (IMV) suitable for adult mechanical ventilation without direct electrification”. It is the opinion of this reviewer that the "primary text" of the manuscript does not at all describe the iterative design that led to the 3D manufacturing of the BubbleVent prototype. On the contrary, the authors seem to overshadow the description of the design methodology, optimization and 3D printing, reporting only part of this iterative design process in the enclosed supplement. In fact, in the supplement, in the section "Development and Manufacturing the BubbleVent", it would have been appropriate to show an exploded view of the final design, in order to identify each single component. Furthermore, the authors state: "Test prints of various BubbleVent parts using other additive manufacturing processes, including vat polymerization and MultiJet Fusion, also demonstrated the suitability of these process to BubbleVent production", however there is no evidence of these results in the paper. Furthermore, the authors declare that: "if components are not available these components can be printed and made airtight", once again the authors do not indicate how to make them airtight. It is the opinion of this reviewer that the supplement should be integrated with the main manuscript, including the sections "Development and Manufacturing the BubbleVent", "Historical development of BubbleVent and different Valve System Iterations", "Design and Production" in section 2 of the main manuscript. While the section "Design Risks and limitations" move it to section 4 of the main manuscript. These are all information that justify the objectives and the qualification of the study, they cannot be postponed to the supplement. Furthermore, the authors should have provided detailed technical information of the printing process such as the infill used for each component, the printing time, the amount of material (specially to justify the low cost they indicated). It is the opinion of this reviewer that the supplement should be integrated with the main manuscript, including the sections "Development and Manufacturing the BubbleVent", "Historical development of BubbleVent and different Valve System Iterations", "Design and Production" in section 2 of the main manuscript. While the section "Design Risks and limitations" move it to section 4 of the main manuscript. These are all information that justify the objectives and the qualification of the study, they cannot be postponed to the supplement. Furthermore, the authors should have provided detailed technical information of the printing process such as the infill used for each component, the printing time, the amount of material (specially to justify the low cost they indicated). In various parts of the text the authors declare that the proposed design can be easily adapted with components available on the market, with semblant interface changes, but they do not clarify which interface and how to modify it. At least one case of practical use could have been demonstrated, the most common for example. The paper is interesting and it is written in well English form but the structure and the presentation of the methodology is not well organized and explained. For these reasons it is opinion of this reviewer the the manuscript needs major revision to be published in this prestigious journal.  

Reviewer 3 Report

There are several issues in the manuscript that should be addressed before further consideration for publication.

  1. Suggest the authors to highlight the changes/improvement made in the new design as compared to commercially available systems.
  2. Some of the parts are 3D printed, however, the details of 3D printing are not provided. For example, what are the designs that are 3D printed? What materials and techniques are used?
  3. 3 test protocols are used. However, are there any replicates within each test protocol? How accurate/repeatable are the results? 
